# A NLP Pipeline for the Automatic Extraction of a Complete Microorganism’s Picture from Microbiological Notes

**DOI:** 10.3390/jpm12091424

**Published:** 2022-08-31

**Authors:** Sara Mora, Jacopo Attene, Roberta Gazzarata, Daniele Roberto Giacobbe, Bernd Blobel, Giustino Parruti, Mauro Giacomini

**Affiliations:** 1Department of Informatics, Bioengineering, Robotics and System Engineering (DIBRIS), University of Genoa, 16145 Genoa, Italy; 2Healthropy, Corso Italia 15/6, 17100 Savona, Italy; 3HL7 Europe, 1000 Brussels, Belgium; 4Infectious Diseases Unit, IRCCS San Martino Polyclinic Hospital, 16132 Genoa, Italy; 5Department of Health Sciences (DISSAL), University of Genoa, 16132 Genoa, Italy; 6Medical Faculty, University of Regensburg, 93053 Regensburg, Germany; 7eHealth Competence Center Bavaria, Deggendorf Institute of Technology, 94469 Deggendorf, Germany; 8First Medical Faculty, Charles University Prague, 12800 Prague, Czech Republic; 9Department of Infectious Diseases, AUSL Pescara, 65124 Pescara, Italy

**Keywords:** hospital-acquired infections, international coding system, laboratory information systems, natural language processing, information extraction

## Abstract

The Italian “Istituto Superiore di Sanità” (ISS) identifies hospital-acquired infections (HAIs) as the most frequent and serious complications in healthcare. HAIs constitute a real health emergency and, therefore, require decisive action from both local and national health organizations. Information about the causative microorganisms of HAIs is obtained from the results of microbiological cultures of specimens collected from infected body sites, but microorganisms’ names are sometimes reported only in the notes field of the culture reports. The objective of our work was to build a NLP-based pipeline for the automatic information extraction from the notes of microbiological culture reports. We analyzed a sample composed of 499 texts of notes extracted from 1 month of anonymized laboratory referral. First, our system filtered texts in order to remove nonmeaningful sentences. Thereafter, it correctly extracted all the microorganisms’ names according to the expert’s labels and linked them to a set of very important metadata such as the translations into national/international vocabularies and standard definitions. As the major result of our pipeline, the system extracts a complete picture of the microorganism.

## 1. Introduction

The recent COVID-19 pandemic highlighted even more the worrying and widespread increasing circulation of pathogenic microorganisms in hospitals, sheltering for elderly, and assisted residences. The Italian “Istituto Superiore di Sanità” (ISS) [1] identifies hospital-acquired infections (HAI) as the most frequent and serious complications of healthcare. A possible definition of HAI is “infections that first appear 48 h or more after hospital admission or no later than 30 days after discharge following inpatient care” [2]. HAIs constitute a real health emergency and require decisive action from both local and national health organizations. The main objective is to build stable and automatic systems dedicated to reporting and epidemiological surveillance. When a pathogenic microorganism responsible for HAI is identified, the information can allow targeted antibiotic treatment, as well as the prompt adoption of specific infection-control measures for multidrug-resistant (MDR) bacteria.

This information is obtained from microbiological cultures of specimens collected from infected body sites, and the outcomes are usually reported in the results of laboratory exams [3,4]. MDR bacteria are defined as those organisms that show resistance to one or more agents in at least three antimicrobial categories during in vitro antimicrobial susceptibility tests [5,6].

MDR organisms are a world health problem causing about 33,000 deaths per year in Europe [7] and 35,000 in the United States [8]. It becomes even more serious when the infection affects critically ill patients, e.g., those admitted to intensive care units (ICUs) [9,10], because it is associated with increased mortality [11]. However, this is not only a human-related problem, because MDR bacteria are frequently found in the environment [12], especially in intensive farming both in agriculture (livestock and poultry) [13,14,15] and in aquaculture [16,17,18].

Although for more than 20 years modern laboratory information systems (LISs) [19,20] managed laboratory analyses, individual centers created their own vocabulary. This is mainly due to the fact that the development of information systems has been non-simultaneous and strongly localized. The resulting need to make results coming out of the single laboratories comparable led to the development of international coding systems and standards devised for the management of terminologies. An example of an international vocabulary applicable in this field is the Logical Observations Identifiers Names and Codes (LOINC). The Common Terminology Service Release 2 (CTS2) [21] is a standard dedicated to terminology management, whose specifications result from the collaboration between the Object Management Group (OMG) [22] and Health Level 7 (HL7) [23].

One of the fields where computerized systems faced many problems was the management of microbiology. This resulted from the high variability of the discipline and the strict link to the habits of individual laboratories (i.e., the coding system on which the nomenclature of bacteria relies, how sensitivity analyses are performed, etc.). Therefore, a contrast arose between the need for more variability and the mandatory use of LISs, imposed by national laws, whose structure in some specific cases may be considered too stiff. Therefore, because of the lack of ad hoc and appropriate fields for representing and managing microbiological-related information, the staff instead preferred exploiting clinical notes written as natural text. Accordingly, clinical notes became an important source of information for biomedical research, patient management, and care, while the necessity of manual inspection made their use expensive in terms of personnel effort and time. The limitations of data/information collection in LIS, on the one hand, and the advantage of concept representations using domain-specific languages instead of data/information representation, on the other hand, are discussed in more detail in Section 4.

This kind of problem can be addressed using artificial intelligence (AI) tools, especially with natural language processing (NLP). It is a branch of computer science that deals with the processing of natural human language by computers, studying the problems connected to the learning, understanding and automatic generation of human language both in written and in spoken form [24,25,26].

The objective of our work was to build a NLP-based pipeline for automatic information extraction from the notes of microbiological culture reports.

This could represent a first step toward the development of a system able to monitor antibiotic prescriptions at a hospital and territorial level in the Abruzzo Region [27].

This paper is an extension of work originally presented at the 18th International Conference on Wearable, Micro, and Nano Technologies for Personalized Health (pHealth 2021) titled “A NLP pipeline for the automatic extraction of microorganisms names from microbiological notes” [28]. The extended version addresses the problem of managing the national and international terminology systems linked to the project and of filtering clinical notes in order to exclude nonsignificant sentences.

## 2. Materials and Methods

The complete schema of the developed pipeline is presented in Figure 1, and it can be divided into four main sections (discussed in detail below). “Data preparation” prepares the inputs values, explaining how we build the vocabulary database, and it loads and preprocesses the clinical notes. “Pattern recognition” takes as input the textual data, converting them into a numerical representation, and it builds three ML-based models that predict if the specific pattern is contained or not in a clinical note. “Pattern removal” takes as input the sentences identified by the model as containing the specific pattern and removes it. “Microorganism detection” takes as input both the filtered and the nonfiltered sentences and first performs the “Genus extension” process before extracting from the clinical notes meaningful information such as the microorganism name and if it belongs to the group of dangerous microorganisms.

### 2.1. Characteristics of the Sample

The collected sample was derived from the LIS of the main hospital of Pescara in Abruzzo Region and was obtained from clinical notes extracted from a 1 month collection of anonymized laboratory referral. It was composed of 499 texts from culture reports, classified as follows:276 (55.3%) texts containing the name of a microorganism where an expert from the hospital confirmed its presence;56 (11.2%) texts needing to be filtered because they contained a pattern that is not useful for our analysis and was, thus, removed. An example of a sentence belonging to that pattern is the following: “Integration of the preliminary report sent on …”. Indeed, we considered the use of synonyms, e.g., “provisional” instead of “preliminary”, and the presence of orthographic errors, e.g., missing letters. Therefore, we decided not to use regular expressions alone as first attempt.

We are waiting for the authorization from Abruzzo region to access the entire LIS system at a regional level to massively test the proposed system with notes produced by a wide range of persons.

### 2.2. Environment and Libraries

We completely developed the pipeline in Python language, and we used the Jupyter Notebook environment. The Python libraries used within this project were as follows:

Pandas: This is an open-source Python library containing data analysis and manipulation tools [29].

Pyodbc: This is an open-source module developed in Python that allows accessing databases through the ODBC (Open DataBase Connectivity).

Natural Language Toolkit (NLTK): This is a worldwide used library to perform text analysis in multiple languages; therefore, it is popular in both academia and research [30]. Some of the operations supported by the NLTK are tokenization, stemming, part of speech tagging (POS tagging), and disambiguation.

SpaCy: This is an open-source library for NLP in Python supporting different languages [31].

Re: This is a Python module that provides operations useful to work with regular expressions [32].

Scikit-learn: This is an open-source library that contains several machine learning algorithms, e.g., classification, regression, and clustering [33].

Seaborn, matplotlib: These are libraries used to produce graphics [34,35].

FuzzyWuzzy: This is a Python library used to manage the comparisons between strings. In detail, it is used to compute the distance between two strings with the same number of characters or not, taking into account the order of words and the allowed maximum frequency of a string. This comparison is based on the Levenshtein distance.
$$lev_{a,b}\left(i,j\right)=\{\begin{array}{c} \begin{array}{cc} \max\left(i,j\right) & \;\;\;\;\;\;\;\;\;\;\;\;if\min\left(i,j\right)=0, \end{array}\\ \begin{array}{ll} \min\{\begin{array}{l} lev_{a,b}\left(i-1,j\right)+1\\ lev_{a,b}\left(i,j-1\right)+1\\ lev_{a,b}\left(i-1,j-1\right)+1_{\left(a_{i}\neq b_{i}\right)} \end{array} & \;\;otherwise \end{array} \end{array}$$
where *i* and *j* constitute the indices of the last character of the two substrings [36].

### 2.3. Data Preparation (I)

Vocabulary building (I.I): We built a vocabulary containing the names of microorganisms (bacteria, fungi, yeasts, and viruses) from the “National Healthcare Safety Network organism list”, including the current taxonomic subdivision which was proposed by Carl Woase in 1990. We mapped the microorganism’s genus and specie into three standard coding systems, at national and/or international level: Italian Clinical Microbiologists Association (AMCLI), Systematized Nomenclature of Medicine—Clinical Terms (SNOMED-CT), and National Healthcare Safety Network (NHSN). Together with the name of the microorganism, we retrieved other metadata, such as the microorganism’s definitions according to Medical Subject Headings (MSH) and the National Cancer Institute (NCI). We stored all the information in an SQL Server database, and we loaded them using the pyodbc tool.

Data acquisition (I.II): We used the pandas library to import data.

Data cleaning, tokenization, and stopword removal (I.III): We cleaned the clinical notes first by removing punctuation and substituting patterns that could be dates with the word “data”. Then, we divided them into minimal text units of analysis, which we called tokens. Then, we proceeded with stopword removal, but considering only words longer than one character in order to exclude from the analysis strings belonging to the class of prepositions, articles, and adverbs, while keeping single letters that could be the abbreviation of a genus name.

### 2.4. Pattern Recognition (II)

Once we loaded and cleaned data, we needed to convert text into a numerical representation that could be used as input for ML algorithms, and we adopted the bag of words technique. This choice was guided by the structure of the sentences that was fragmentary and did not respect any strict syntactic rules. Therefore, we preferred to use a context-free representation.

Numerical representation building (II.I): Bag of words (BoW) is a numerical representation of text that describes the occurrence of words within a document. It involves two main issues: a vocabulary of known words (or n-grams of characters) and a measure of their presence in the text. BoW representation does not keep any information about the structure or order of words in the document. The possibility to add grouped words (called n-grams of words) to the vocabulary allows capturing a little of the meaning from the document. The resulting numerical representation was composed of both n-grams of characters and n-grams of words following the proportion of 70:30. We decided to select more features composed by n-grams of characters in order to deal with misspellings, abbreviations, and limited syntactic rules. We tested the model performance considering 10 possible total numbers of selected features from 10 to 100 with a step size of 10. We obtained the best performance with a total number of features equal to 90.

Classification (II.II): We used the aforementioned numerical representation to learn a supervised binary classifier to predict whether the observed pattern was present in the clinical note or not. Specifically, we compared the performances of three classifiers, as described below.

SVM is a supervised learning method that can be used to perform classification analysis on both linear and nonlinear data [37]. The main aim of SVM is to find a line or a hyperplane that maximizes the distance between the classes (support vectors) when placed between them [38]. If data are not linearly separable, then they can be transformed using a kernel function from a low-dimensional to a high-dimensional structure to make the data separable.

Logistic regression (LR) is a method of statistical analysis used to estimate the relationship between a dependent variable and at least one independent variable, minimizing the Euclidean distance between the true label and the model output. Specifically, for binary classification, the output variable is modeled by a sigmoid ranging between 0 and 1. We introduced model sparsity adding an L_1_ penalty term [39].

Random forest (RF) is an ensemble of K decision tree classifiers created from a different bootstrap sample. The trees are built by sampling a random subset of the attributes at each internal node in the decision tree. The random sampling of the attributes reduces the correlation between the trees in the ensemble [40].

We split the dataset into a learning and a testing set with the proportion of 80:20. On the learning set, we performed the hyperparameter search through a tenfold cross-validation, which iteratively split the learning set into a training and validation set. They were respectively used to learn the model with all the possible combinations of hyperparameters and to evaluate the performances thereafter. Then, we learned the three models with the selected set of best hyperparameters, and we evaluated the model performances on the testing set. We repeated the classification 20 times, shuffling the data each time. In order to guarantee reproducibility of results, we set the random state equal to the loop index.

Models Evaluation (II.III): We evaluated the performance of the three ML models in terms of accuracy.

### 2.5. Pattern Removal (III)

Once the algorithm classified the clinical notes as “containing”/“not containing” the pattern, we used regular expressions to remove the uninformative pattern from the identified notes.

The schema of the regular expression was as follows: \b[Ii]\w.+?\bdata\b.

The elements of the expression are defined below.

\b asserts the position of a word boundary. In this case, we want the pattern beginning with ‘I’ (the first letter of the word ‘Integrazione’ (integration), which can be abbreviated and/or can be uppercase or lowercase in the notes).

\w matches any word character and ends with ‘data’ (the word that we substituted for all dates in the data cleaning phase).

. matches any character (e.g., letters, numbers, and punctuation) except for line terminators.

+? matches the previous token between one and unlimited times, the fewest times possible, but expanding as needed.

### 2.6. Microorganism Detection (IV)

Genus Extension (IV.I): We stored the microorganism names using the binomial nomenclature originating from the Linnaeus classification [41]. It is composed of two terms: the first is the genus name with the first letter capitalized; the second is the species name in lower case. Usually, after a microorganism’s name is written once in a text, then it can be substituted with its first capital letter, followed by a period, in subsequent mentions. However, considering the brevity of the clinical notes, a shared agreement is to always write the abbreviated form, despite the entire genus having not yet been introduced. This binomial nomenclature does not allow the use of an abbreviation composed of two letters for the genus. Nevertheless, even though the microorganisms should be written according to this strict rule, we decided to keep words composed of only one character and not to use a regular expression, because we considered that abbreviations may be not written correctly, e.g., by using abbreviations that are not followed by a period, or where uppercase letters are followed by a period but not followed by lowercase letters. Hence, we performed the extension of the microorganism genus. Specifically, we compared the “*n* + 1” token with each species of the vocabulary. If the similarity index between the two tokens was greater than or equal to 98, then we checked the token *n*”. If the token “*n*” began with the same letter of the genus of the species in position “*n* + 1”, we substituted the token “*n*” with the genus name found in the vocabulary. The schema of the treatment is presented in Figure 2.

Microorganism name extraction (IV.II): Initially, we tried to carry out a lexical and morphological analysis, but the lack of morphological structure of the clinical notes resulted in no good results. Therefore, we extracted the microorganism name by comparing each token “*n*” of the preprocessed clinical notes and the genera in the vocabulary using the FuzzyWuzzy library. The complete workflow of the microorganism name extraction phase is shown in Figure 3.

In particular, considering the genus extraction, we set the threshold of the similarity index at 75, while we set the threshold of the species index as 85 (as they were typically written correctly).

Other Information Extraction (IV.III): Together with the identification of the genus and species, in order to highlight microorganisms that could be potentially dangerous, we also searched the clinical notes for the keyword “alert”, which is an explicit indication of microbiologists regarding the danger of the identified microorganism. Similarly, but much less frequently, the “non-alert” bi-gram, with which the microbiologists indicate the harmlessness of the microorganism, may be present. To address both cases, we performed a search at the token level for the keyword “alert”. If identified at the n position, we checked if token *n* − 1 matched the negation “non”.

## 3. Results

### 3.1. Identification and Removal of a Specific Pattern

In the process of information extraction from the microbiological notes, it is useful to identify nonmeaningful sentences, e.g., “integration of the provisional report of …”. The lack of morphological structure in the sentences led us to use a count-based method to build a numerical representation of the clinical notes.

Figure 4 summarizes the mean values of accuracy obtained using the three classifiers over the 20 iterations per each total number of features, shuffling the data each time.

We obtained best results in terms of mean accuracy across classifiers (99.06%) with a total number of features equal to 90. The SVM classifier with a Gaussian kernel obtained a mean accuracy of 98.99%, logistic regression obtained a mean accuracy of 98.99%, and random forest obtained a mean accuracy of 99.19%.

This means that the pattern was correctly identified using all classifiers, and it can be securely removed from the specific clinical notes.

### 3.2. Genus Extension

Our sample of clinical notes contained a total number of 107 abbreviated genera followed by their species. After the system elaboration process of the notes, all 107 genera were extended, and they completely matched with the expert indications.

### 3.3. Microorganism Detection

Our sample of clinical notes was composed of 499 texts, and 276 (55.3%) of them actually presented the name of a microorganism. We performed two tests.

First, we introduced the entire sample into the module for microorganism extraction.

Then, we introduced only those notes that actually contained the name of the microorganisms.

The system correctly identified all microorganism names in both cases. In detail, it found 416 genera, and, as shown in Figure 5a, the majority of them (321) had a similarity index of 100. This was also a consequence of the genus extension process.

Figure 5b shows that ‘*Staphylococcus*’ was the genus name with the lowest score; in particular, it usually presented a very low similarity index, between 76 and 80, if a species was not specified. Indeed, we frequently found not only the strictly scientific term, but also the Italian term in the notes, because *Staphylococcus* is among the most widespread bacteria and is frequently mentioned in the common discourse.

This behavior affected the similarity index; in particular, *Staphylococcus* and ‘stafilococco/stafilococchi’ (Italian terms referring to the *Staphylococcus* genus) have 14 and 12 letters, respectively, within which only nine coincide, representing a Levenshtein distance of 5 (i.e., five changes are needed to transform the first word into the other). On the other hand, species never showed a Wuzzy index lower than 88.

Lastly, we introduced a weight, which was a decimal parameter ranging between 0 and 1. It could be associated with the genus–species couple or only with the genus, if present alone. As the same word (genus and/or species) could be associated with more than one genus/species, this process was necessary to highlight the maximum Wuzzy indices. An example of the system output is shown in Table 1; the similarity index of the two genera *Acinetobacter* and *Acetobacter* was 92, which is quite high. Thus, in order to identify the correct genus without any doubt, we compared the following token with all species of that specific genus present in the vocabulary. If a match was found (with a Wuzzy index over 98), then we assigned to that genus–species couple a weight parameter equal to 1, while the others received a weight of 0.

Otherwise, if the clinical note did not contain any species, and the two genera that could correspond to the same word had an identical Wuzzy index, e.g., as a consequence of a spelling error, then the algorithm would assign to both genera an equal weight of 0.5.

### 3.4. Other Information Extraction

The whole sample included 48 clinical notes that contained the keyword “alert”. Our algorithm was able to correctly discriminate between the notes that contained the bi-gram “non-alert” (*N* = 9) and those that contained the keyword alone (*N* = 39).

## 4. Discussion

In general, the pattern recognition and the genus extension phases led to good results. The first achieved a mean accuracy value of 99.06% considering all three classifiers, while the second extracted all the names of microorganisms reported by the experts from the hospital. We should consider, however, that some ambiguities could be found during this second phase. Indeed, there are a few microorganisms with identical species and whose genera begin with the same letter. If one such case appears, then the system will duplicate the clinical note and it will extract both microorganisms, but both notes will be associated with a weight equal to 0.5. However, we should specify that, luckily, these kinds of ambiguities are quite rare. A well-known example is the *intermedius* species, which can belong to both the *Staphylococcus* genus and the *Streptococcus* genus. *Staphylococcus intermedius* is quite frequent in animals; however, it is reported as a human pathogen in very few cases, most of which are associated with exposure to animals, especially dogs. On the contrary, *Streptococcus intermedius* is one of the major causes of brain abscesses, but very few cases of this condition are documented annually in Italy, with an incidence that is less than 0.1% per year. Therefore, we can affirm that the probability that such ambiguity is present in the report notes of the microbiological laboratory is extremely rare.

The major result of our pipeline is that we can extract a wider picture of the microorganism, because each microorganism is stored together with other metadata in the build vocabulary, such as the definition according to MeSH and its translation into national and international vocabularies. Furthermore, the pipeline also extracts the property of the microorganism under healthcare surveillance. Therefore, we can say that the system returns an object with its main characteristics. Once we accurately describe the microorganism, we can consider its identification in the clinical note as a trigger event of a series of messages and communications in accordance with the management policies of resistant microorganisms. Thus, it is possible to build a path to safeguard the patient and the community against the resistant microorganism [42]. The above-described system should be integrated in a multidisciplinary context. Correctly integrating objects from any viewpoint of the system in question requires its formal representation and management using the ISO 23903 Interoperability and Integration Reference Architecture [43]. ISO 23903 standardizes a model and framework for representing any type of system from the perspectives of the involved domains, its architectural composition/decomposition, and the related development process of implementable information and communications technology (ICT) solutions.

A limitation of the presented work is the low number of samples considered due to the fact that, to be delivered to researchers outside the laboratory, all these notes were checked individually and manually in order to avoid the illicit dissemination of personal data. In the near future, the correct structuring of electronic health records (which enables in constitutive law the reuse of clinical data for the purposes of scientific research) and greater awareness of the health risk that antibiotic-resistant bacteria constitute will result in a much higher number of notes to be analyzed. The more important methodological limitations of our project and ways to overcome them are discussed in the next section.

## 5. Future Work—Challenges and Solutions

Collecting, as well as storing and retrieving, representational objects of facts, systems, and processes is always a matter of the language and related grammar used to perform those actions. A simpler and more constrained language and ruling grammar, which is equivalent to the expressivity of languages/ontologies, facilitates processing of the outcome. However, highly expressive languages are less complete. This is a crucial challenge of knowledge representation in any business system including health and its special domains such as microbiology or infectious diseases.

Any business system can be represented using ICT ontologies. This holds for data stored in LIS databases, information models to represent the system’s objects, and business process models representing the business processes needed to meet the intended business objectives. However, the justification of correctness and completeness of structure and behavior of the represented ecosystem can only be provided from the ecosystem’s business view using the involved domains’ ontologies. Justification of structure and behavior representation includes the representational components, their underlying concepts, their relations, and the related constraints. Figure 6 illustrates the related business system according to ISO 23903. The domains involved are clinical domains, managing patient care, and supporting facilities such as laboratories and microbiological departments to provide diagnostic services, as well as regulatory domains (policy domains) such as legal affairs, administration, security, and privacy management. Within the development process, the real-world system is then transformed into the different viewpoints of the intended ICT solution from the business process modeling (enterprise view) through the informational representation of all entities involved (information view) up to implementable artifacts (engineering view) and their management (technology view). The views in that order are represented by languages/grammars with increasing constraints and decreasing generative power, as well as decidability.

The technology view and engineering view are represented by data, while the computational view and information view are represented by information using related presentation styles such as programming languages or UML, respectively. The enterprise view represents the enterprise knowledge using business process modeling languages, and the business view represents the domain knowledge using domain ontologies (Figure 7). The different levels in the model hierarchy allow for different actions necessary for designing and running the business process according to Krogstie [45].

For performing process-related actions, the enterprise view is necessary. For taking strategic and operational decisions, driving innovations, and enabling comprehensive collaboration, the representation of the business system in its comprehensive context using the ontologies of the directly and indirectly involved domains, guided by top-level ontologies according to ISO 21838 [47], is inevitable. In other words, the taxonomies used to analyze the business system must be replaced by ontologies, thereby not just considering the domain knowledge, but also the knowledge space in question. This involves not only the naming of entities, but also the underlying concepts and comprehensive relations. More details on the challenges and solutions can be found, e.g., in [48] or [49], as well as in the introductory paper to this volume.

## 6. Conclusions

The main aim of this work was to develop an NLP pipeline to support the automatic extraction of the microorganisms’ names and important information contained in microbiological notes of culture reports. We decided to preprocess the notes before the extraction process by removing not meaningful sentences, such as “integration of the provisional report of…”. We performed this task by applying machine learning methods to the numerical representation of the texts obtained using the bag of words technique. All the microorganisms present were extracted correctly; hence, the main goal was achieved. Lastly, considering that our vocabulary is based on international nomenclature standards, the presented pipeline can be applied to similar laboratory notes from other hospitals across the nation.

A next step of this project will be to also automatically extract from the same clinical notes the antibiogram data. In particular, key information that should be considered is the sensitivity of the specific microorganism to each single antibiotic tested. In the context of healthcare transformation toward pHealth or even 5P medicine (personalized, preventive, predictive, participative, and precision medicine), we have to advance the system further.

## Figures and Tables

**Figure 1 jpm-12-01424-f001:**
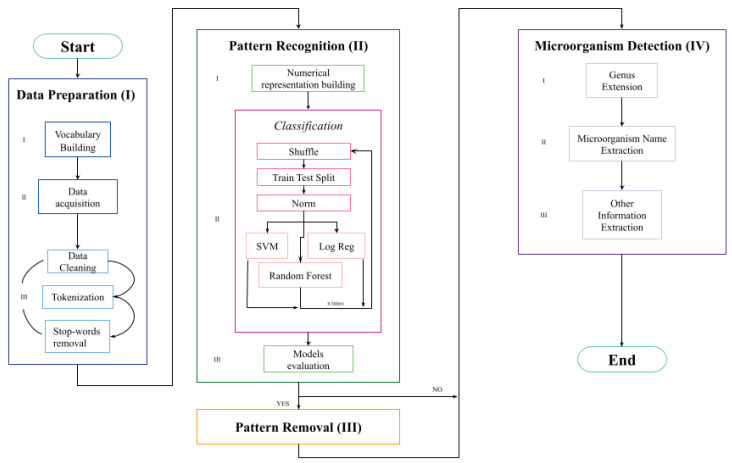
Complete schema of the pipeline. It can be divided into 4 main sections: data preparation, pattern recognition and removal, and microorganism detection.

**Figure 2 jpm-12-01424-f002:**
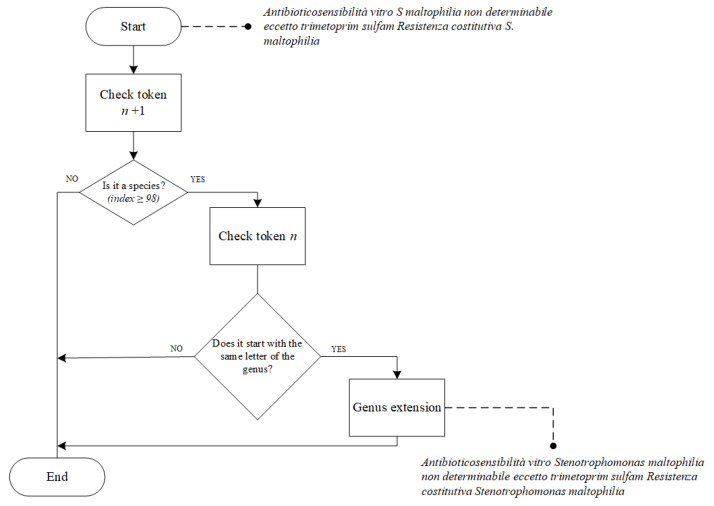
Genus extension decision flow. The figure also includes an example. In the upper part of the figure, there is a sentence already preprocessed but before the genus extension phase, whereas, in the lower part, we can see the extended version. First, the maltophilia species is identified, while “S” as the first letter of genus Stenotrophomonas is extended.

**Figure 3 jpm-12-01424-f003:**
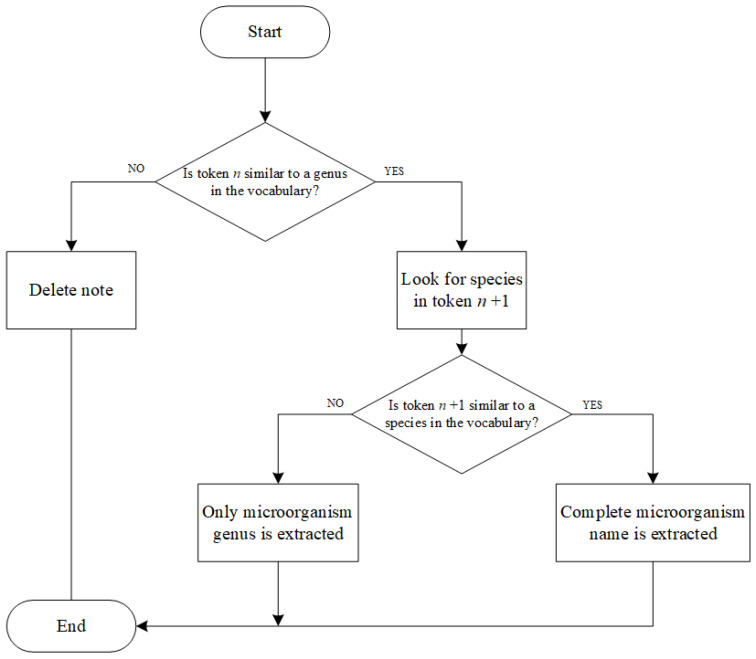
Microorganism name extraction decision flow. In contrast to the genus extension phase, in this process, the token “*n*” is first considered, and it is compared to the genera listed in the vocabulary. Only if a match is found (with a high similarity index) is the token “*n* + 1” considered and compared to the list of species in the vocabulary that belong to the identified genus.

**Figure 4 jpm-12-01424-f004:**
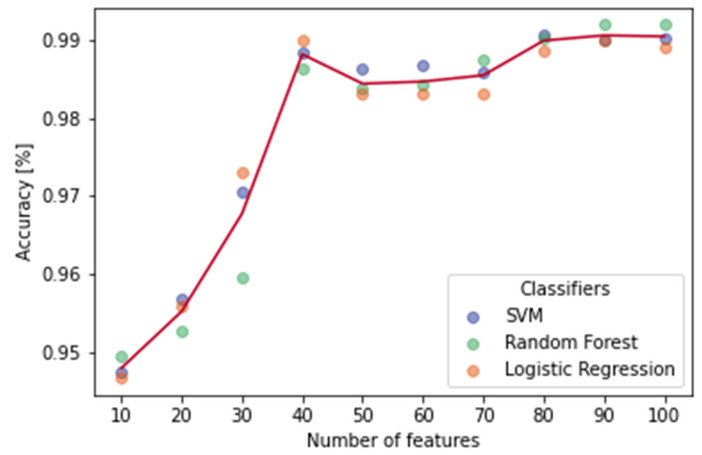
Mean accuracy performances of the three classifiers displayed for each value of the total number of features. Each data point is the mean value of 20 values obtained by shuffling the data.

**Figure 5 jpm-12-01424-f005:**
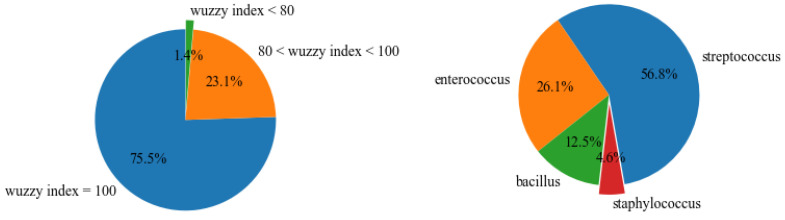
System performance for genus extraction. (**a**) Similarity index percentage distribution of genera. (**b**) Percentage distribution of genera found with low indices.

**Figure 6 jpm-12-01424-f006:**
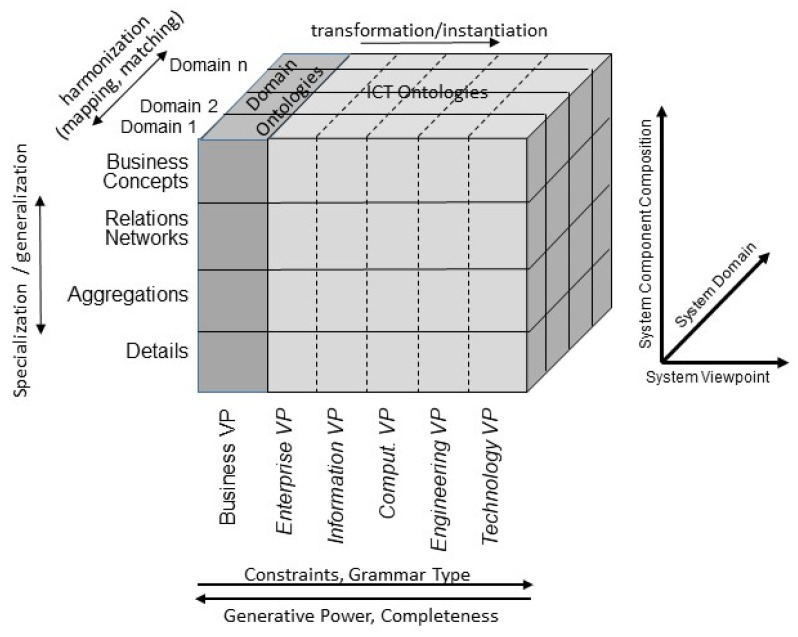
Generic business system representation according to ISO 23903, including the language/grammar characterization according to Chomsky [44].

**Figure 7 jpm-12-01424-f007:**
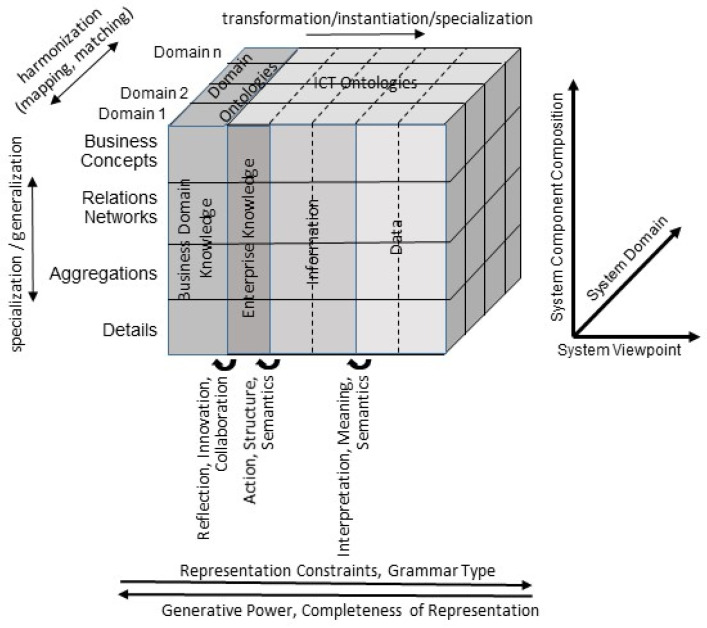
Generic business system representation according to ISO 23903 from the perspective of the knowledge pyramid after Aamodt and Nygard [46] and the model hierarchy after Krogstie [45].

**Table 1 jpm-12-01424-t001:** Example of the system output returned when the input was “Gram negativi profilo proteomico riferibile ad *A baumannii*. Propensione di *A baumannii* alla pan-resistenza eccetto colistina (Microorganismo alert)”. The displayed columns correspond to the genus from the vocabulary, the specific word or character in text which the genus matches to, the genus Wuzzy similarity index, the species from the vocabulary, the word in text which the species matches to, the species Wuzzy similarity index, the clinical note divided into tokens, and the weight parameter.

Genus	Match Genus	Wuzzy Index Genus	Species	Match Species	Wuzzy Index Species	Clinical Notes	Weight
*Acinetobacter*	*A*	100.0	*baumannii*	*baumannii*	100.0	[94,‘propensione‘,‘NaN‘,‘Acinetobacter‘,‘b…	1
*Acetobacter*	*A*	92.0	NaN	NaN	NaN	[94,‘propensione‘,‘NaN‘,‘Acinetobacter‘,‘b…	0
*Aminobacter*	*A*	83.0	NaN	NaN	NaN	[94,‘propensione‘,‘NaN‘,‘Acinetobacter‘,‘b…	0
*Citrobacter*	*A*	83.0	NaN	NaN	NaN	[94,‘propensione‘,‘NaN‘,‘Acinetobacter‘,‘b…	0

## Data Availability

Data used in this project are available as Appendix A.

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
