# Peer review of "A NLP Pipeline for the Automatic Extraction of a Complete Microorganism’s Picture from Microbiological Notes"

_jpm, 2022, doi:10.3390/jpm12091424_

Round 1

Reviewer 1 Report

The authors have proposed a NLP pipeline for automatic extraction of microorganisms picture from microbiological notes. Though there are some work has been done but it requires major changes.

1. The presentation is very weak, it must be improved

2. There are some serious grammar and punctuation mistakes, it is suggested to proofread the native English speaker

3. The methodology section should be rewritten covering all the aspects of automatic extraction

4. Recommended to add some latest journal citations.

5. Some more graphical or tabular information for various evaluation metrics can be added.

Author Response

We changed section “Materials and Methods” describing in detail all the elements mentioned in Figure 1. We changed Figure 1 as well by inserting some indexes in order to support the reader in the understanding of information management process. We cited these indexes in the text to ease the reading of that specific section. We added Figure 3 to support the understanding of the microorganism name extraction process.

Finally, one of our collaborators, that is an expert mother tongue in English, revised the whole text.

Reviewer 2 Report

The authors' work on developing an NLP pipeline for the extraction of hospital-acquired infections (HAIs) is timely and relevant to the current healthcare need. I have the following suggestions for the manuscript.

Page 5 of 14, Line 189: Please mention if you considered using an external dataset for validation

Page 8 of 14, Line 279: Figure 4 is low quality and hard to read. Please re-do the figure with larger fonts and high quality

Page 8 of 14, Line 299: Figure 5 is low quality and hard to read. Please re-do the figure with larger fonts and high quality

Page 10 of 14, Line 381: Figure 6 is low quality and hard to read. Please re-do the figure with larger fonts and high quality

Page 11 of 14, Line 391: Figure 7 is low quality and hard to read. Please re-do the figure with larger fonts and high quality

Page 12 of 14, Line 428: It would be great if the authors could also make for code available. 

Author Response

Page 5 of 14, Line 189: Please mention if you considered using an external dataset for validation.

We are waiting for region Abruzzo approval to access the entire regional LIS. This will give us the opportunity to obtain a bigger sample of data that will be comparable but written from different healthcare professional in different laboratories and so it could be considered as an external validation of our system. This way we could hopefully demonstrate the robustness of our system.

Figure quality issue

We made results presentation clearer by improving figures quality and font dimension.

Page 12 of 14, Line 428: It would be great if the authors could also make for code available.

At this point of the project code is composed by several parts that require a certain interaction with the user. After we will obtain the authorizations that we are waiting for, we want to implement a system that will be better integrated and automatized. We will evaluate, together with the company, which we are collaborating with, the possibility to made code available.